# Towards a Counting Point Detector for Nanosecond Coherent X-ray Science

**Sooheyong Lee [1,2,*], Hyusang Kwon [1], Byeong-Gwan Cho [1] and Eric C. Landahl [3]**

[1]   Korea Research Institute of Standards and Science, Daejeon 305-340, Korea
[2]   Department of Nanoscience, University of Science and Technology (UST), Daejeon 305-340, Korea
[3]   Physics Department, DePaul University, Chicago, IL 60614, USA
*   Correspondence: sooheyong@gmail.com

**Abstract:** We present the technical realization of a high-speed hard X-ray single-photon counting-detection scheme based on a commercial avalanche silicon photodiode and high-speed oscilloscope. The development is motivated by the need to perform pulse-resolved photon-correlation and pump-probe studies at synchrotron sources with densely packed pulse patterns that result in high repetition rate pulses on the order of hundreds of MHz. Commissioning experiments are performed at the 1C PAL-KRISS beamline at PLS-II of South Korea operating at a burst mode maximum repetition rate of 500 MHz. In such a high count-rate measurement, detector dead-time can lead to a distortion of counting statistics. We are able to model the counting behavior of our detector under these conditions with a detector dead-time comparable to time between X-ray pulses, implying that nanosecond X-ray photon correlation spectroscopy should be possible at diffraction-limited light sources.

**Keywords:** time-resolved X-ray scattering; single photon counting; dense-fill pattern; dead-time correction; diffraction limited light source

## 1. Introduction

Synchrotron sources have become an indispensable tool for probing complex processes in versatile choices of condensed matter and chemical and biological materials. During the past two decades, aided by highly collimated X-ray beams available at 3rd-generation light sources, notable breakthroughs have been made in measurement techniques that exploit a large transversely coherent flux, such as X-ray photon correlation spectroscopy (XPCS), coherent diffraction imaging [1–3], diffraction microscopy [4], and ptychography [5]. The pulsed nature of the X-ray sources has also enabled time-domain science to understand the structures and dynamics of transient systems that had been previously considered inaccessible [6,7]. For instance, recent studies reported measuring coherent X-ray diffraction patterns (speckles) from atomic-scale ordering in amorphous systems, such as liquid and glasses [8–10], which opens up possibilities for measuring dynamics at the relevant length scales [11–15]. In time-domain methods such as XPCS, the intensity fluctuations of these speckles reflect the change in the sample dynamics, of which characteristic time scales can be extracted from intensity-to-intensity time correlation functions. However, typical detectors that are used for such measurements carry an intrinsic frame rate of kHz, which defines the experimental time resolution limit. Therefore, the time-correlation experiments today are mostly limited to studying model systems with slow dynamics. Ever-increasing demands to measure smaller and faster features in fundamental science and industrial applications have instigated the development of high brightness and transverse coherence of the beam beyond what is available at current storage-ring based sources.

The recent emergence of low-emittance storage ring light sources based on a multi-bend achromat lattice, known as fourth-generation storage rings (4GSR), deliver orders of magnitude more brilliant X-ray radiation than that previously available from 3rd-generation light sources [16–19]. The high spatial coherence of the beam promises the possibility to

probe fast microscopic dynamics via X-ray photon correlation spectroscopy. However, such light sources are expected to employ a very dense-fill pattern (electron pulse structure) to reduce the charge density per electron pulse. As a result, they are expected to operate at very high repetition rates of hundreds of MHz, which allows only a few nanoseconds between subsequent X-ray pulses that inevitably carry much fewer X-ray photons per pulse as compared to 3rd-generation light sources. Ultimately, the successful implementation of time-resolved measurements such as XPCS or stroboscopic (pump-probe) diffraction at next-generation light sources requires both the sufficiently fast detection of individual X-ray pulses and the correct analysis of registered X-ray photons [20,21]. Previously, such a task has been enabled by using state-of-art mechanical choppers or the high-speed electronic gating of one-dimensional [22,23] or two-dimensional detectors [24,25]. Here, the use of mechanical choppers is not a viable option as they only operate at the limited opening times of hundreds of nanoseconds [26,27]. On the other hand, the use of electronic gating requires a detection scheme that is capable of recovering its active state before the arrival of the subsequent X-ray pulses. These recoveries typically take several tens of nanoseconds and are even slower for hard X-rays due to the long X-ray penetration depth into the sensors. Therefore, time-resolved measurements using dense fill patterns would be extremely difficult to implement, if not impossible, using current detector technology. A potential solution for these challenges is collecting X-rays via single photon counting with appropriate dead-time corrections.

Fundamental excitation and subsequent relaxation processes in condensed matter systems evolve at very fast speeds. Observing them directly will become possible with next-generation X-ray sources. Here, we show that detectors and acquisition systems also exist that can match their anticipated performance. In this work, we demonstrate pulse-resolved hard X-ray photon detection capable of handling the 500 MHz repetition rate of incoming X-rays, which is equivalent to the fill pattern of the upcoming 4GSR in South Korea [19]. The high-speed single-photon detection scheme was implemented at PLS-II of Pohang Accelerator Laboratory.

## 2. Experimental Setup

Time-resolved measurements at synchrotrons utilize the pulsed nature of X-ray radiation that originates from the time structure of electron pulses that circulate around the storage ring. Typically, a series of multiple electron pulses is filled into consecutive radiofrequency (rf) buckets that are evenly spaced. As shown in Figure 1, our demonstration was performed at the 1C PAL-KRISS beamline at PLS-II, which generates hard X-ray pulses with down to 2 ns pulse spacing and approximately 50 ps (FWHM) in pulse duration. The 2 ns minimum spacing is set by the frequency of the storage ring rf cavities of 500 MHz. The storage ring revolution period is 1 μs, corresponding to 500 rf buckets per revolution. The monochromatized X-ray beam with the central wavelength of 10 keV and an energy bandwidth of (dE/E) of $10^{-4}$ is delivered from cryogenically cooled double Silicon (111) crystals to the detector that is positioned downstream of aluminum foil filters (See Figure 1). In our experiment, we make use of combinations of multiple aluminum filters with varying thicknesses to control the transmitted X-ray intensity. The thickness of the filters in the measurement were 4, 8, 16, 32, 64, 128, 256, 512, 1024, and 2048 μm. The transmitted intensities at different filter settings were calibrated by using ion chambers before and after the attenuators. Using these filter settings, we are able to perform the experiment with less than 0.2 percent of the initial intensity. Here, a crucial component of this detection scheme is the capability of resolving a single X-ray pulse. For the X-ray detection, we used the Menlo systems optical avalanche photodiode (APD-210, 0.5 mm detector size) with its front glass windows removed to increase detection efficiency. Real-time, single-shot oscilloscope-trace measurement provides an accurate way to monitor X-ray photon counting. The signal of the APD is fed to a high-bandwidth oscilloscope (LeCroy WaveRunner 604Zi) that is triggered by a 1 MHz rf-revolution clock. Given the manufacturer's specified intrinsic rise time of 500 ps for the APD and the bandwidth (BW) of 400 MHz for the oscilloscope that

translates to a rise time of $\frac{0.35}{BW} = 875$ ps, we calculate an effective rise time of our acquisition system to be $RT_{eff} = \sqrt{(875\,\text{ps})^2 + (500\,\text{ps})^2} = 1.007$ ns. This value is consistent with our result shown in Figure 1, where the direct X-ray beam is attenuated using calibrated filters so that the detector receives less than a single photon event within 2 ns time-interval. Figure 1 (inset) shows the oscilloscope trace of a single X-ray pulse collected by the APD, which displays a full recovery within 2 ns time-interval. Assuming that the rise and fall time responses are the same as shown in Figure 1, we estimate the dead time of 2 ns. We note that a single X-ray photon can arrive at the detector at any time during the 50 ps duration of an electron pulse.

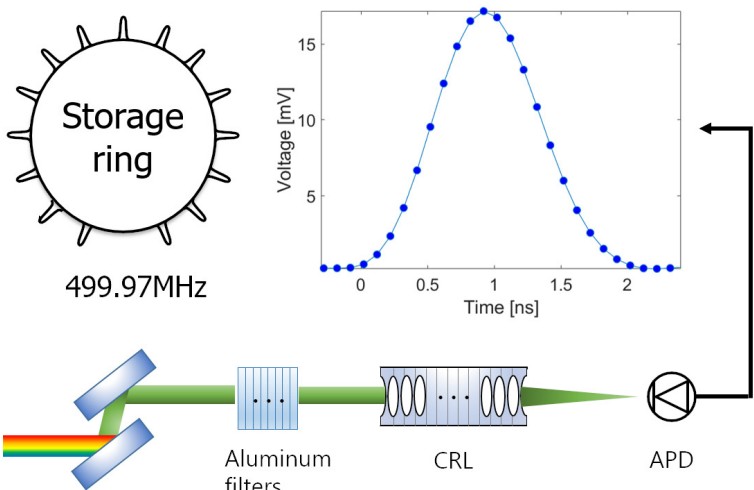

**Figure 1.** Sketch of the data acquisition scheme. 10 keV X-ray radiation is delivered from the Silicon (111) monochromator and focused on the 0.5 mm APD sensor by using a set of compound refractive lenses (CRL). The X-ray flux for the measurement is controlled by using varying combinations of aluminum filters positioned before the CRL. The waveforms are acquired by feeding the APD signal to the oscilloscope.

## 3. Results: High-Speed Data Acquisitions

Under the PLS-II special operation mode, 300 electron pulses are injected into the consecutive rf-buckets while the rest of the remaining 200 buckets are unfilled. Figure 2a shows an oscilloscope trace of the X-ray pulses that are collected by the APD within 10 revolution clock cycles. The signal from the X-ray pulses striking the detectors is manifested as positive voltage peaks where the average pulse height corresponding to the detection of an individual 10 keV X-ray photon is 8.2 mV. Here, the peaks with an amplitude greater than 2.5 mV are identified with red circles and are counted as the registered photon events in post-processing. Figure 2b shows a histogram of the time gap between two successive X-ray pulses. This result demonstrates that the detector and acquisition system has the ability to resolve the timing of a single X-ray pulse at the count rate of 34 MHz, which corresponds to approximately one detected X-ray photon for every ten pulses. As shown in Figure 3, re-binning the histogram into individual buckets approximates the arrival time of the next photon as a continuous variable. The exponential distribution result implies that the photons are uncorrelated. This is expected since there is no coherent scattering (speckle) in this measurement.

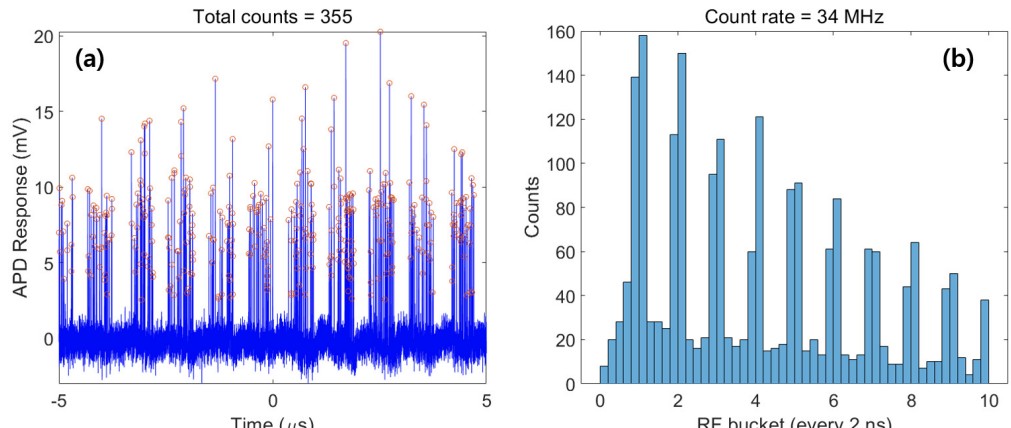

**Figure 2.** (**a**) Waveform traces from the APD recorded from 10 revolution clock cycles. Peaks with the amplitude exceeding the threshold of 2.5 mV are recorded as X-ray photons and marked by red circles. (**b**) Histogram of time between two successive pulses.

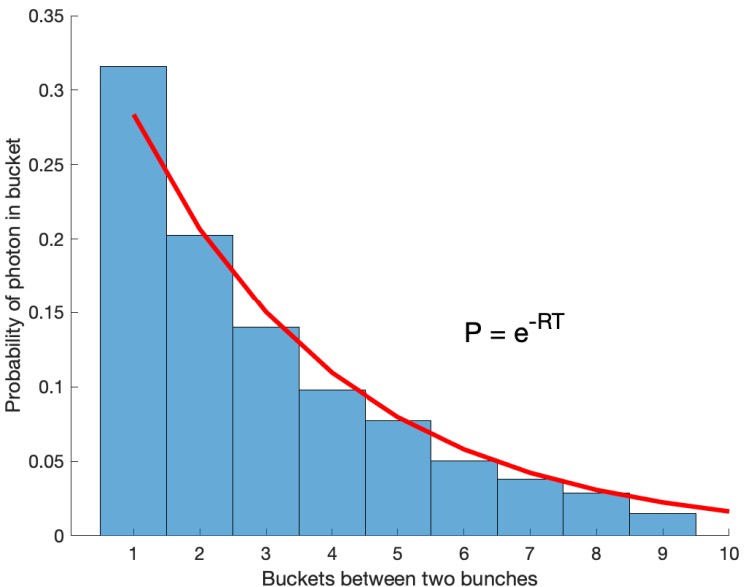

**Figure 3.** A binned probability distribution of photons per rf-bucket, which approximates the exponential distribution $P = e^{-RT}$ of a continuous random variable where $R$ is the detected photon rate (34 MHz in this example) and $T$ is the time between pulses, or 2 ns.

## 4. Discussion: High-Dynamic Range Measurement

Noise-free photon counting over a wide dynamic range is demonstrated by repeating the oscilloscope data acquisition and off-line peak-finding analysis for different combinations of aluminum filters placed before the detector. The uncertainty in the observed count rate $N_o$ is calculated based on Poission statistics where the random error in the number of counts is $\sqrt{N_o t_{obs}}$, where $t_{obs}$ is the observation time, equal to one hundred round trips of the storage ring (100 µs, or ten successive oscilloscope traces such as shown in Figure 2). At a relatively low flux, the detector count rate response is linear, as shown in Figure 4. However, at higher count rates, a nonlinearity occurs because there is no pulse-height analysis of the output signal, and therefore multiple photon hits in one pulse cannot be distinguished from a single photon [28,29].

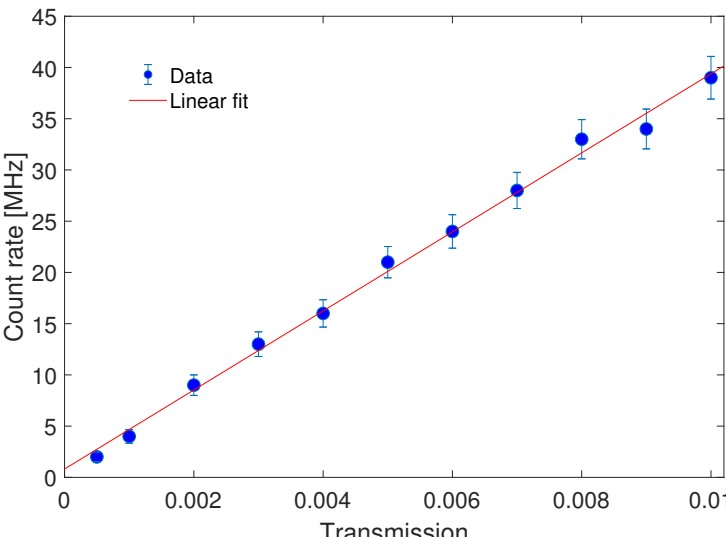

**Figure 4.** Correlation between the APD count rate and X-ray transmission after the aluminum filters at relatively low count rates.

The most important criterion for such pulse counting at a high photon flux is the minimum resolvable time separation between two adjacent events. In most high-speed applications, the limiting time arising from the combined effect of the detector response and the associated electronics is called the dead-time, which ultimately determines the temporal resolution of the detection scheme. For correctly recovering unbiased counting statistics from measured data, one needs to compensate for the electronics and detector dead-times. For several decades, many researchers have proposed models to correct for the detection dead-times [28,30–32]. Here, we explore two different dead-time models. First, we consider the *isolated dead-time model* that is applicable for a detector that is not limited by the pulse structure, i.e., it has a dead-time $\tau_{iso}$ less than the pulse spacing ($T$), and the only difference between the true ($N_t$) and observed ($N_o$) count rates are a result of the probability of multiple photons arriving within a single pulse, which can not be distinguished from a single photon observation,

$$N_t = -\frac{1}{\tau_{iso}} ln(1 - N_o \tau_{iso}). \tag{1}$$

In Figure 5, this model (dashed red line) is applied to APD counts measured over a wider range of upstream X-ray filter transmission. The saturation behavior yields a dead-time of $2.17 \pm 0.10$ ns, which is slightly larger than the actual pulse separation time of $T = 2.00$ ns. The discrepancy between the model and the saturation data indicates that we are just beyond the applicable range of the isolated model, where the pulse repetition rate has no effect on detector performance. Next, we compare the saturation curve to the *non-extended dead-time model*, which allows for the detector to be rendered inoperative for a fixed time $\tau_{nex}$ after each observed count,

$$N_t = \frac{N_o}{1 - \tau_{nex} N_o}. \tag{2}$$

This model is shown in Figure 5 (dot-dashed green line) and shows a better agreement with the saturation curve data than the isolated model, indicating that the detector recovery time has a measurable but minor and correctable affect on the observed count rate at 500 MHz. The non-extended dead-time obtained from the nonlinear curve fit of the data is $1.90 \pm 0.04$ ns, just below the pulse separation time. Here, to analyze the high count-rate data sets shown in Figure 5, we perform a nonlinear curve fitting using Equation (1) on data and obtained an error of 0.1 ns. We note that the error bars in Figure 5 are too small to be seen due to the high count rate.

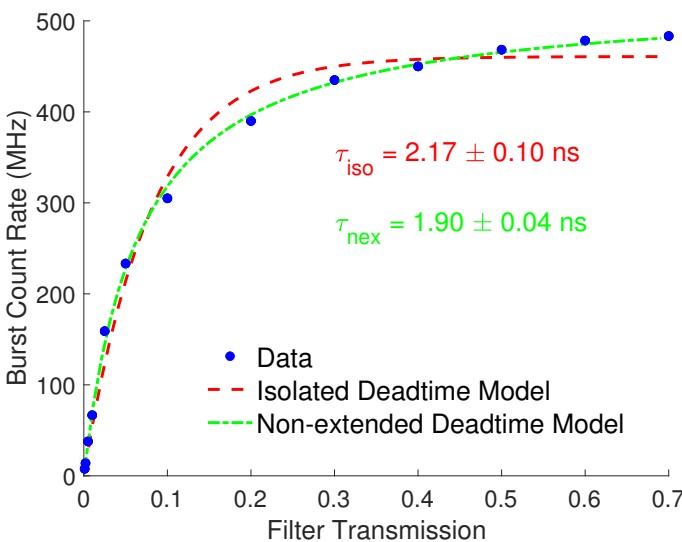

**Figure 5.** The observed APD count rate saturation that occurs as aluminum X-ray transmission filters are removed from in front of the detector is fitted to both an isolated dead-time model (Equation (1)) and a non-extended dead-time model (Equation (2)). Both models indicate the detection system dead-time is comparable to the minimum X-ray pulses separation of 2.00 ns. The error bars from counting statistics in the observed count rate are less than 1% for observed count rates above 100 MHz and too small to be seenn on this plot.

We note that the hard X-ray single photon detection here is performed via a silicon-based APD with a small sensor size of 0.5 mm, which may be appropriate for X-ray speckle measurements in 4GSR beamlines in the ultra-small-angle X-ray scattering regime. Smaller APDs are available with a faster response time that could be appropriate for high-coherence sources used on samples with high X-ray damage resistance. On the other hand, the 0.5 mm size is inconvenient for most pump-probe diffraction studies, and therefore we also tested a prototype high-speed X-ray APD (UAPD, Enertrex Corp., Crestwood, IL, USA) with a considerably larger sensor size of 10 mm and a higher X-ray detection efficiency due to a thicker absorption depth. The UPAD displayed a comparable temporal response in its rise time [33]. However, the unit was not used in this work due to its relatively slow recovery response; this detail should be resolved in our upcoming work. In the future, we also intend to use a high-speed discriminator coupled to an FPGA-based counting board capable of clocking at 500 MHz that is synchronized to the the storage ring frequency to collect the data instead of taking the full wave-form record. This effort will reduce the data load and increase the acquisition efficiency.

## 5. Conclusions

We have demonstrated the capability of utilizing photon counting of X-ray pulses at the full repetition rate of the PLS-II storage ring source. The high X-ray flux available at synchrotrons, especially operating in dense fill mode, can cause saturation of photon-counting, which results in distortions in the measured signals. We use combinations of high-speed APD and the analysis of dead-time effects to make full use of the dynamic range of the detector scheme. Our approach is applicable to any single-event counting measurements with electronics and detector dead times. Thus, the correction scheme will find wide application in all time-resolved photon counting experiments at upcoming light sources where high transverse coherence and a very dense electron fill pattern are expected, such as 4GSR.

Finally, we note that the optimal count rate is approximately $T/2$ for a detector working in the isolated dead-time regime ([34]); above this counting rate (250 MHz in this case), the dominating source of count rate uncertainty is no longer counting statistics but

instead becomes the under-counting of X-rays when multiple photons arrive in a single pulse. For application to XPCS, achieving isolated dead-time is extremely important since single-pulse dead-time correction only results in a correction to the true count rate and not a correction to temporal correlations, unlike detectors that experience a recovery time longer than $T$. The ability of 4GSR sources to provide these high count rates in real samples of interest at relevant scattering angles is yet to be determined; however, we anticipate that single-point counting detectors such as the one demonstrated here will be able to handle this task and potentially permit the measurement of atomic length-scale fluctuations at atomic time-scales.

**Author Contributions:** Conceptualization, S.L. and E.C.L.; methodology, all authors; analysis, S.L. and E.C.L.; writing–original draft preparation, S.L. and E.C.L.; and writing–review and editing, all authors. All authors have read and agreed to the published version of the manuscript.

**Funding:** E.C.L. was supported by a DePaul CSH FSRG. This research used resources of the Advanced Photon Source, Sector 7ID-C, a U.S. Department of Energy (DOE) Office of Science User Facility operated for the DOE Office of Science by Argonne National Laboratory under contract No. DE-AC02-06CH11357. This research was supported by the National Research Foundation of Korea under contract No. NRF-2019K1A3A7A09033397.

**Institutional Review Board Statement:** Not applicable.

**Informed Consent Statement:** Not applicable.

**Data Availability Statement:** The data presented in the study are available on request from the corresponding author.

**Acknowledgments:** We acknowledge useful discussions regarding APD development with Steve Ross of Enertrex Corporation and on the topic of detector response time correction with Jesús Pando of DePaul University. Experiments at 1C PAL-KRISS beamline at PLS-II were supported in part by MSIT and POSTECH.

**Conflicts of Interest:** The authors declare no conflict of interest.

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
