# Peer review of "Towards a Counting Point Detector for Nanosecond Coherent X-ray Science"

_applsci, doi:10.3390/app12178886_

Round 1
Reviewer 1 Report
The authors present clearly the design and test results of a counting point detector for nanosecond coherent x-ray science, which was tested at a high count rate up to 500 MHz and represented with an isolated dead-time model. It sounds and can be published with three minor modifications,
-Line 69 It should be better to give the parameters of APD-210, for example response time or rise time, etc. (also the used oscilloscope, such as bandwidth).
-Line 98 Could you estimate the dead time considering the APD-210 and oscilloscope?
-Line 104 How did you get the values of T and its error? And it seems that you did not give the errors of measured count rate in Figure 5 but in Figure 4 you gave.
Reviewer 2 Report
The manuscript describes technical realization of a high-speed hard x-ray single photon counting detection scheme designed for working with high intensity X-ray beams and, in particular, with synchrotron beams. The issue of detecting high-brilliance beams in state-of-art X-ray light sources is very topical since modern detecting systems time resolution do not allow direct measurements with such intensive beams. Therefore, various approaches for the indirect measurements are of interest to scientists and technicians dealing with such sources.
Unfortunately, I do not fully understand the idea of the described work. First, the manuscript seems to me more like a technical report than a scientific article. Besides, I would like to clarify if I understand correctly, that the main idea of the proposed technique is to combine commercial silicon photodiode and high-speed oscilloscope with aluminum filtration system, which decrease X-ray beam intensity up to detector time resolution? Are authors use any special algorithm to extract data from amplitude measurements and transform it to single-photon measurement? I would say that this manuscript needs more technical details and careful description of the research design. Besides, there are few minor remarks can be done according to the text of the manuscript:
- Figures 4, 5. What means “Filter transmission” scale? Is this a ration of beam intensity after/before aluminum filter? Does it mean that authors made measurements with less than 0.2 percent of initial intensity? What was the thickness of filters in the measurements?
- What is authors’ model for counting behavior of the detector? Is this a simple equation between lines 107 and 108?
- Did authors consider an influence of incoherent scattered X-ray in aluminum filters that can be significant for thick filters?
- Figure 5. Are experimental data and simulated curve in a good agreement? It seems to me that a few points are significantly differs from the curve. I believe the figure would benefit from measurement errors.
- Line 113. What is USAXS regime?
- Please, check the language of the manuscript one more time.
I believe that the manuscript needs major revision. In my opinion, it would benefit from more details described in the text.
Reviewer 3 Report
The manuscript describes a setup capable of detecting hard x rays with correlations on a ns timescale, suitable for new 4th generation synchrotron sources with high repetition rate pulse structure. The paper is concise clearly written and of obvious interest to researches in the field.
For those not familiar with the condensed matter techniques referenced, it would be helpful to include brief details of what the minimum dead time requirement would be and why, and this relates to a confusing array of terms, which could be made clearer.
Line 75. Single x ray pulse, is confusing, the pulses are 50ps wide, is this the detector output profile in response to a single photon detection event? Are bunches and pulses the same thing, i think that an insert with the timing diagram of the source, would be helpful in Fig 1 with clarification of pulses, bunches buckets etc.
Minor corrections, questions
Fig 1 feeding not feed
Fig 2 caption
Should be peaks with greater than the detection threshold?
Round 2
Reviewer 2 Report
The manuscript describes technical realization of a high-speed hard x-ray single photon counting detection scheme designed for working with high intensity X-ray beams and, in particular, with synchrotron beams. The issue of detecting high-brilliance beams in state-of-art X-ray light sources is very topical since modern detecting systems time resolution do not allow direct measurements with such intensive beams. Therefore, various approaches for the indirect measurements are of interest to scientists and technicians dealing with such sources.
I believe the manuscript can be accepted for publication in present form.